

# A multiscale ocean data assimilation approach combining spatial and spectral localisation

Ann-Sophie Tissier[1], Jean-Michel Brankart[1], Charles-Emmanuel Testut[2], Giovanni Ruggiero[2], Emmanuel Cosme[1], and Pierre Brasseur[1]

[1]Univ. Grenoble Alpes, CNRS, IRD, Grenoble INP, IGE, Grenoble, France
[2]Mercator Océan, Toulouse, France

**Correspondence:** Ann-Sophie Tissier (annsophie.tissier@gmail.com)

**Abstract.** Ocean data assimilation systems encompass a wide range of scales that are difficult to control simultaneously using partial observation networks. All scales are not observable by all observation systems which is not easily taken into account in current ocean operational systems. The main reason for this difficulty is that the error covariance matrices are usually assumed to be local (e.g. using a localization algorithm in ensemble data assimilation systems), so that the large scale patterns are

removed from the error statistics.

To better exploit the observational information available for all scales in CMEMS assimilation systems, we investigate a new method to introduce scale separation in the assimilation scheme.

The method is based on a spectral transformation of the assimilation problem and consists in carrying out the analysis with spectral localisation for the large scales and spatial localisation for the residual scales. The target is to improve the observational

update of the large scale components of the signal by an explicit observational constraint applied directly on the large scales, and to restrict the use of spatial localisation to the small scale components of the signal.

To evaluate our method, twin experiments are carried out with synthetic altimetry observations (simulating the JASON tracks), assimilated in a $1/4°$ model configuration of the North Atlantic and the Nordic Seas.

Results show that the transformation to the spectral space and the spectral localization provides consistent ensemble esti-

mates of the state of the system (in the spectral space,or after backward transformation to the spatial space). Combined with spatial localisation for the residual scales, the new scheme is able to provide a reliable ensemble update for all scales, with improved accuracy for the large scale; and the performance of the system can be checked explicitly and separately for all scales in the assimilation system.

## 1  Introduction

Over the last decades, the spectral window of the oceanic processes observed from space has steadily increased. At the same time, model resolution has also improved to better understand and interpret the observed signals. This progress in observations and models is a challenge for ensemble data assimilation. Because the size of the ensemble is always very small compared to the number of degrees of freedom to be monitored. The model is usually too expensive to perform large-size ensemble simulations.



This means that the probability distribution of the possible states of ocean is described by a small sample as compared to the dimension of the subspace over which uncertainties develop. In particular, the rank of the ensemble covariance matrix is much smaller than the rank of the real error covariance matrix. A traditional approximation to solve this problem is to localise this error covariance matrix (Houtekamer and Mitchell, 1998; Hamill et al., 2001; Testut et al., 2003; Brankart et al., 2011). The

analysis is then applied locally by only using observations within a defined radius of influence which is bound to decrease with the broadening of the spectral window controlled by the data assimilation. The control of large scales, namely larger than this radius of influence, thus results from the combination of a large number of local analyses.

The large scales structures, although they are well observed (in the ocean by altimetry, ARGO floats,...), are therefore only indirectly controlled by the algorithm. Observations contain simultaneously information about small scale structures (especially

at the observation point) and about larger scale structures, taken into account the full observational network. Spatial localisation does not directly take advantage of each scale contained in the observations system.

Because of the limited size of the ensemble, it is difficult to explicitly control the full range of scales without separating the spectral components of the signal. Separation of scales during the analysis step of data assimilation algorithms allows us to adjust localisation according to the considered spectral band of the signal. This is helpful to directly control the large

scales which are frequently and precisely observed (altimetry, ARGO floats, ...). To separate scales in data assimilation, two approaches have been previously studied : the multiscale filter and the spectral transformation. The multiscale filter consists in separating the signal in two spectral bands, delimited by a cutting scale, in order to achieve two distinct ensemble analysis in the spatial space (Zhou et al., 2008; Zhang et al., 2009). In this scheme, applied to an EnKF, two distinct localisation windows are used to exploit correlations over a longer distance for the analysis of the large scales. A more approximate version has also

been proposed by simply combining the increments obtained for each of the two spectral bands (Miyoshi and Kondo, 2013). A comparable approach was proposed for 3DVar systems by Li et al. (2015). Alternatively, Buehner (2012) proposed a spectral transformation approach within an EnVar system, which is a spatial and a spectral localisation with a wavelet transform. This method is more generic because scales are separated continuously from the largest scales to the smallest scales. Localisation is used to neglect the correlations between the components of the signal which are distant both in terms of spatial location

and in terms of scales. However, this method would be expensive for large systems and could be difficult to insert in a global ocean assimilation system. More recently, Buehner and Shlyaeva (2015) and Caron and Buehner (2018) have developed a new formulation of this algorithm for EnVar systems. It incorporates the multiscale filter idea of decomposing the signal in several spectral bands and it avoids the compute removal of the between-scale covariances (Buehner and Charron, 2007). This formulation makes use of an augmented spatial/spectral ensemble covariance matrix, whereas the result of the analysis is still

computed in the spatial space.

Following a similar idea of combining the multiscale filter and spectral transformation approaches, we propose in the present paper to combine these two algorithms by applying a spectral analysis with spectral localisation (hereinafter called spectral localisation) to the large scales components of the signal and a spatial analysis with spatial localisation (hereinafter called spatial localisation) for the residual scales. By separating the components, we avoid using an augmented covariance matrix and

we thus potentially neglect useful statistical relationships. However, this makes the multiscale system less expensive and easier



to implement in an existing ensemble data assimilation system. It is indeed expected that the spectral transformation of the large scales is cheap enough to be applied to large size global ocean system, and that spectral localisation is more appropriate than spatial localisation to capture the large scale components of the observed signal. On the other hand, for the small scale components, the spectral transformation becomes too expensive, and the local correlation structure prevails. The target is thus to improve the observational update of the large scale components of the signal by an explicit observational constraint applied directly on the large scales, and to restrict the use of spatial localisation to the residual scale components of the signal. These analyses should be done one after the other to be included in an existing sequential algorithm as operated for instance by Mercator Ocean.

The performance of this multiscale observational update is then studied with an example application in the context of CMEMS systems. We performed a 70-member ensemble simulation using the oceanic model NEMO (Nucleus for European Modelling of the Ocean (Madec, 2008)) version 3.6 at $1/4°$ with the CREG4 configuration (Dupont et al., 2015) as part of the CMEMS project. This configuration of the North Atlantic and Nordic Seas is currently used at Mercator Ocean for developing and testing the future assimilation system. This configuration is thus appropriate to check that our new algorithm can be integrated in the data assimilation system of Mercator Ocean (SAM2) used for the CMEMS program.

The objective of this paper is to describe the multiscale observational update algorithm that we have developed and to evaluate its performance using the CREG4 ensemble system. The paper is organised as follows. In Sect. 2, we present the practical problem that we want to solve : we describe the prior ensemble, the observation system, and the difficulties associated to the multiscale correlation structure. In Sect. 3, we present the spectral transformation that is applied in this study to better display the multiscale correlation structure. In Sect. 4, we present the algorithm that we have developed to make a better use of the correlation structure for all scales. This combines spectral localisation for the large scales with spatial localisation for the small scales. In Sect. 5, we evaluate the resulting algorithm using the application problem described in Sect. 2. This is done by studying the reliability and resolution of the updated ensemble for each wavelength of the control variables.

## 2  Application problem

The purpose of this section is to introduce the example application that is used in this paper to study the performance of the multiscale observational update. This example application is chosen to serve the development of the CMEMS systems and to display the multivariate character of the assimilation problem. The model configuration and the prior ensemble simulation are described in Sect. 2.1; the assimilation problem is described in Sect. 2.2; and the multiscale character of the ensemble correlation structure is described in Sect. 2.3.

### 2.1  Ensemble model simulation

Our example application is based on a $1/4°$ resolution model of the North Atlantic and the Nordic Seas. We used the oceanic model NEMO (Nucleus for European Modelling of the Ocean, (Madec, 2008)) version 3.6, with the CREG4 configuration as part of the CMEMS project. NEMO, used by a large community, is developed by European institutes and is used by the



majority of the CMEMS stakeholders. It is a primitive equation model which computes the following prognostic variables :
3D velocities, sea surface height, salinity and temperature. ERA-Interim reanalysis data, produced at ECMWF (Dee et al.,
2011), are used for the atmospheric forcing. CREG4 is a realistic configuration for the North Atlantic and the Nordic Seas, at
the $1/4°$ horizontal resolution. This configuration is described in the work of Dupont et al. (2015), except that we use a $1/4°$
version instead of the nominal $1/12°$ resolution. It has been developed by Environment Canada and coupled with Mercator
Ocean's SAM2 data assimilation system. The aim was to build a realistic description of the mean state and variability in the
Artic Oceans and adjacent seas. The CREG4 configuration is currently used by Mercator Ocean and its resolution is sufficient
to evaluate the multiscale assimilation algorithm, therefore we use it for our study.

   Uncertainties in the model are explicitly simulated using the standard NEMO stochastic parametrisation module developed
by Brankart et al. (2015). The aim is to produce an ensemble with a sufficient spread for all variables, especially for the
observed variables, all over the domain. This technique has been used to simulate 6 different kinds of uncertainties in the
model as described in Table 1. Concerning the equation of state, we used the stochastic parameterisation proposed in Brankart
(2013). The two standard deviation values given in the table corresponds to the standard deviation of the random walks in
the horizontal and the vertical directions. Uncertainties in air-sea fluxes uncertainties are parameterised using a multiplicative
noise (with gamma pdf to make it positive) applied to the turbulent exchange coefficients simulated by NEMO following the
algorithm from Large and Yeager (2009) extended to other parameters (and evaluated by Mercator Ocean). Ice uncertainties
are parameterised using stochastic processes representing uncertainties in ice strength, in ice albedo, in ice/sea and ice/air drag
coefficients.

   With this stochastic modelling system, a 70-member ensemble simulation, without assimilation, is performed for the 8-
month period between mid-January and mid-September 2011. It will be used to performed the analyses in the present paper.
This ensemble simulation yields a probability distribution for the evolution of the system, in particular the ensemble mean,
hereafter $\langle x_f \rangle$ where $\langle \cdot \rangle$ indicates an ensemble mean over the members of the ensemble, and the background error covariance
matrix of the prior ensemble, $P_f$. Figures 1a and 1b show respectively the ensemble mean ($\langle x_f \rangle$) and the standard deviation of
the prior ensemble. This ensemble is appropriate to illustrate the multiscale analysis in our study. Indeed, as will be shown later,
the spread of the ensemble spans a wide range of scales from basin scale to mesoscale. Large scales as well as small structures
are well represented which will allow us to evaluate our separation scale algorithm. Nevertheless, the standard deviation is too
small in the regions of strong eddy activity as the Gulf Stream. This is mainly due to the model configuration (CREG4) which
causes an excessive dissipation of the turbulent kinetic energy. Moreover, the variability is too large close to east Siberian sea.
However, these characteristics do not affect the evaluation of the multiscale algorithm that is performed in this study.

## 2.2   Definition of the twin experiments

The assimilation problem investigated in this study is based on twin experiments with altimetry. In this kind of experiment,
the true state is known and synthetic observations are built from this true state. It is generated by the same model to which
data assimilation is applied. This method has the advantage that the effectiveness of the different algorithms can be directly
evaluated thanks to the known true state. One member of the ensemble simulation is left apart to be used as a reference (the





simulated truth) from which the observations will be simulated : $x_{\text{true}}$. The prior ensemble (indicated thereafter by the subscript $f$ as forecast during an assimilation step) used in the experiments is thus a 69-member ensemble.

In this study, to illustrate the behaviour of the multiscale algorithm, we will concentrate on studying the observational update of the prior ensemble on August 30, 2011. Figure 1c shows the true anomaly : $x_{\text{true}} - \langle x_f \rangle$. This true anomaly will be

used as a reference to evaluate the effectiveness of the different localisation schemes and of the multiscale analysis. Synthetic observations are simulated by adding a simulated observational noise ($\epsilon$ : a Gaussian noise with $5$ cm standard deviation) to the true state ($x_{true}$) with the observation operator ($H$) along the track of the JASON altimeter :

$$y^o = H x_{\text{true}} + \epsilon \tag{1}$$

In this experiment, a 10-day observation window is chosen to have the best coverage provided by JASON. Figure 1d shows the

resulting synthetic observation of the SSH. JASON altimeter does not provide any observation above $66°$ North latitude. In our example, there is thus no available observation to correct, during the analysis step, the large ensemble variance observed close to east Siberian sea (Fig. 1b). This is something that the multiscale approach will have to cope with.

The observational update of the prior ensemble will be performed with a square-root algorithm. The analysis scheme used at Mercator-Ocean is derived from the Singular Evolutive Extended Kalman filter (SEEK) (Pham et al., 1998; Brasseur and

Verron, 2006) which is very close to whose used by the Ensemble Transform Kalman Filter (ETKF) (Bishop et al., 2001). It can be applied indifferently in spatial space as well as in spectral space. In the multiscale algorithm developed in this study, nothing will be changed in the core of the square root algorithm : the only novelty is that a spectral transformation is applied before the observational update to allow spectral localisation rather than spatial localisation. A spatial localisation scheme has been already developed and evaluated (Testut et al., 2003; Brankart et al., 2011). For this study, it has been adapted to be used

in the spectral domain.

### 2.3 Ensemble correlation structure

The 69-member ensemble correlation structure (without the true state, which has been left apart) is illustrated in Fig. 2a and Fig. 2b. It has been computed according to two arbitrary reference points : one in the Gulf Stream, and the other in the North-East Atlantic Ocean off the coast of Portugal. Ensemble correlation structure shows how the assimilation of an observation at

this reference point will influence the other regions during the analysis step. For $N = 69$ members, the correlation coefficient is significant at $95\%$ if it is larger in absolute value than $0.2367$. In both examples, the most important and significant values, i.e. where the observation will have an impact, are mostly confined around the reference point. Some significant correlations are observed further, but their values are lower and not reliable enough to be used during an assimilation step. The usual solution to avoid the spurious effect of non-significant ensemble correlations is to perform a spatial localisation during the analysis step.

It consists in completing the correlation structure provided by the ensemble by the assumption that only local correlations are significant and usable (Houtekamer and Mitchell, 1998; Hamill et al., 2001). The long range correlations are thus assumed to be zero to perform the analysis step.





However, if we look at the same correlation structure (from the same ensemble) for the large scale component of the signal (characteristic scale larger than $L \approx 187$km), as illustrated in Fig. 2 (bottom panels), we see that there are significant correlations over a much larger range. Hence, a significant information of the large scale signal is available even if the size of the ensemble is small. But these are usually masked by the presence of the small scale signal. In the standard spatial localisation,

these long distance correlations are not used and thus a part of the large scale information is not used during the analysis step. The goal is now to find a way to correctly exploit these correlations in the assimilation scheme to better estimate the large scale signal.

It seems difficult to explicitly control all scales of the system without separating the different spectral components of the signal. In this study, the main idea is to do a spectral transformation of all variables of the system in order to do the analysis in

the spectral space before going back in the spatial space to do the next steps of the assimilation scheme.

## 3 Spectral transformation

The purpose of this section is to describe the linear transformation that will be applied on the state vectors and on the observation vectors to separate scales. The forward and backward transformation of the model data are described in Sect. 3.1 and Sect. 3.2 respectively; the transformation of observations and observation errors is described in Sect. 3.3; and the effect on the ensemble

correlation structure is studied in Sect. 3.4.

### 3.1 Forward transformation : projection on the spherical harmonics

The forward transformation step means transforming into the spectral space each input parameter used for the analysis, namely each member of the prior ensemble, but also observations and observational errors. A full two-dimensional signal in spherical coordinates, $f(\theta, \phi)$, can be projected on spherical harmonics $Y_{lm}(\theta, \phi)$ by the following spectral transformation (ST) :

$$\text{ST}: \quad f_{lm} = \int_{\Omega} f(\theta, \phi) Y_{lm}(\theta, \phi) \, \mathrm{d}\Omega \tag{2}$$

where $l$ and $m$ are the degree and order of each spherical harmonics, with $l \in \mathbb{N}$ and $|m| \leq l$. In principle, the integral in Eq. (2) extends over the whole sphere $\Omega$. However, in the assimilation system, all fields $f(\theta, \phi)$ that need to be transformed are anomalies with respect to the ensemble mean. In practice, it is thus possible to extend $f(\theta, \phi)$ with zeroes outside the available domain ($f(\theta, \phi) = 0$ on continents and outside the model domain) in order to compute the integral over the whole sphere. For

a multivariate three-dimensional variable, this transformation can be applied to each vertical level of each model variable.

This spectral transformation provides a new point of view on the ensemble because it separates scales. Each degree $l$ of the spherical harmonic indeed corresponds to a wavelength of a spherical harmonic $\lambda = \dfrac{2\pi R_c}{l}$ with $R_c$ the Earth radius and thus to a characteristic scale $L = \dfrac{\lambda}{2\pi}$. The reversible spectral transformation preserves the information for all degrees $l \leq l_{\max}$. Thus, the transformed fields contain the same information, until this degree, as that shown in the spatial space in Fig. 1.

Figure 3a shows the standard deviation of the SSH of the prior ensemble in the spectral space until the degree $l_{\max} = 60$,





which corresponds to a wavelength $\lambda \approx 667$ km and a characteristic length $L \approx 106$ km. Figure 3b shows the result of the spectral transformation applied to the true SSH anomaly. Similar patterns have been observed by Wunsch and Stammer (1995) from early altimetric observations. Most of the variability is concentrated at large scales (small $l$). The variance becomes weak for meridional structures, i.e. for $\mid m \mid \to l$.

## 3.2 Backward transformation : scale separation

From the spectrum $f_{lm}$, the full field $f(\theta, \phi)$ can then be reconstructed using the inverse transformation :

$$\text{ST}^{-1}_{l_{\min} \to l_{\max}}: \quad f(\theta, \phi) = \sum_{l=l_{\min}}^{l_{\max}} \sum_{m=-l}^{l} f_{lm} Y_{lm}(\theta, \phi) \tag{3}$$

This inversion can be constrained to specific scales by choosing the values of $l_{\min}$ and $l_{\max}$. This is how the method separates scales.

Any spectral band can thus be extracted by choosing the range $[l_{\min}; l_{\max}]$ appropriately. Figure 4 shows the result of the extraction of the large scales applied to each member of the ensemble and to the true anomaly to keep only the large scales. In this case : $l_{\min} = 0$ and $l_{\max} = 34$, which corresponds to a wavelength $\lambda \approx 1177$ km and a characteristic scale $L \approx 187$ km. Small scales structures have been properly removed and only large scales structures remain visible on the figure.

## 3.3 Transformation of the observations

In theory, transformation of observations is not needed to separate scales in the assimilation system. It should be sufficient to introduce the scale separation operator in the observation operator of the existing algorithm. However, for practical reasons, the algorithm that we are proposing requires a preprocessing of the observations to separate scales. This is done to keep the algorithm easy to implement in an existing system : nothing new needs to be implemented except the scale separation operator, and to keep the resulting algorithm efficient enough to be applicable to large size assimilation system.

In this section, we show how this transformation of observations can be performed by regression of the observations on the spherical harmonics (see Sect. 3.3.1) and how the statistics of the observational errors can be transformed accordingly (see Sect. 3.3.2).

### 3.3.1 Regression of observations

For all observations that are not available on a regular grid (for which Eq. (2) could directly be applied), the spectral transformation can be performed by linear regression of the innovation vector (anomaly of the observations with respect to the ensemble mean) on the spherical harmonics.

The approach is to look for the spectral amplitudes $f_{lm}$ so that the corresponding field $f(\theta, \phi)$ (up to degree $l_{\max}$ following Eq. (3) with $l_{\min} = 0$) minimises the following distance to observations $f^o$ :

$$J^o = \sum_{k=1}^{p} \frac{1}{(\sigma_k^o)^2} [f(\theta_k, \phi_k) - f_k^o]^2 \tag{4}$$





where $p$ is the number of grid points of the domain; $f_k^o$ is the observation at coordinates $(\theta_k, \phi_k)$; $\sigma_k^o$ is typically the observation error standard deviation (including the representativity error corresponding to the signal above degree $l_{\max}$) at coordinates $(\theta_k, \phi_k)$. If the observation system is insufficient to control all spectral components with sufficient accuracy, the final penalty function $J$ can include a regularization term $J^b$ as $J = J^o + \alpha J^b$, where the parameter $\alpha$ can be tuned (between $0$ and $1$) to modify the importance of $J^b$ with respect to $J^o$. The regularization term $J^b$ is the following norm of the spectral amplitudes $f_{lm}$ of Eq. (2) :

$$J^b = \sum_{l=0}^{l_{\max}} \sum_{m=-l}^{l} \frac{f_{lm}^2}{\sigma_{lm}^2} \tag{5}$$

where $\sigma_{lm}$ is typically the standard deviation of the signal along each spherical harmonics.

In practice, several additional modifications may need to be introduced in the algorithm and have been implemented for our study. (i) For a non-global model domain (such as CREG4), it may be better to reduce the basis of the spherical harmonics (for each degree $l$) to the subspace that is effectively spanned by the prior ensemble. (ii) For numerical efficiency reasons, it can be useful to perform the regression locally (over a local range of degree $l$), and then iterate until convergence. (iii) In case of large regions without observations (as the Nordic seas for spatial altimetry), it can be useful to add zero bogus observations to avoid triggering a spurious signal where no observation is available.

### 3.3.2 Observational error

The observational error results from both the initial Gaussian error with a standard deviation of $5$ cm introduced on the true member to create the observation, and also the partial observation coverage and hence the algorithm used to do the regression. In theory, this error can be quantified. We suppose that the observational error is decorrelated at large scales. Indeed, the large scales correlations of this observational error are small compared to the observed large scale structures. This assumption will be verified in Sect. 5.2 by the consistency of the rank histograms. As part of our twin experiment, we propose the following procedure.

This error has been quantified following these steps. In this twin experiment, the true state is known. The chosen true member from which the observation has been created, initially belongs to an ensemble of $N + 1 = 70$ members. Then, in the same ways described above, $N + 1$ true members, $x_{true}^i$ with $i \in [1; N+1]$ can be used to generate observations $y^{o,i}$. It is then possible to evaluate the standard deviation of the observational error in the spectral space, by the RMS between $\mathrm{ST}_{\mathrm{regr}}\left[y^{o,i} - H\langle x_f \rangle\right]$ where the operator $\mathrm{ST}_{\mathrm{regr}}$ provides the spectrum resulting from the regression of innovations $y^{o,i} - H\langle x_f \rangle$, and $\mathrm{ST}\left[x_{\mathrm{true}}^i - \langle x_f \rangle\right]$ where the operator $\mathrm{ST}$ provides the spectrum of the corresponding true anomalies $x_{\mathrm{true}}^i - \langle x_f \rangle$, following Eq. (2).

This method is directly applicable to twin experiments, and can be transposed to a real system by simulating observational error and looking at how it is transformed in the spectral space.





### 3.4 Transformed correlation structure

We need to study the main dependencies and correlations between the different spectral components of the ocean fields in order to determine whether and how the scale separation could be used in the data assimilation scheme. Figure 5 shows two examples of ensemble correlation maps between spectral amplitudes. It is comparable to Fig. 2 but in the spectral space (amplitudes $f_{lm}$

of Eq. (2)). Ensemble correlation structure is computed according to reference points in the spectral space and indicated by crosses in Fig. 5. It shows how the assimilation of the signal of an observation at these reference points will impact the other scales during the spectral analysis step. Similarly to Fig. 2, the significant and maximum area is confined near the reference points. Correlations between very different scales are weak and should be neglected by the data assimilation scheme and reduced to zero. This property allows to introduce the scale separation in the data assimilation scheme with a reasonable cost.

To exploit this property, a spectral analysis thus also requires to be localised, at least for the large scales in our study. The method of spectral localisation is the same as that usually used in the spatial space. For the same reasons, each localisation window will contain a number of degrees of freedom sufficiently low to be controlled with an ensemble of moderate size.

### 4 Combining spatial and spectral localisation

The objective of this section is to introduce and demonstrate the multiscale observational update algorithm, combining spectral

localisation for the large scales and spatial localisation for the small scales. In Sect. 4.1, we show how spectral localisation can be obtained using the spectral transformation presented in Sect. 3, and how it can be combined with spatial localisation to build up the multiscale observational update algorithm. In Sect. 4.2, we compare the spatial and spectral localisation schemes, and demonstrate the improvement brought by spectral localisation in the control of the large scales. In Sect. 4.3, we use this comparison to determine the critical scale, $l_c$, above which spatial localisation starts performing better than spectral localisation.

This critical scale is the key parameter that specifies how spatial and spectral localisation are combined in the multiscale observational update algorithm.

### 4.1 Multiscale observational update algorithm

We propose an algorithm for the multiscale analysis based on a combination of a spectral analysis with spectral localisation for the large scales (described by Eq. (7)) and a spatial analysis with spatial localisation for the residual scales (described

by Eq. (6)). The large scales are defined by the critical scale $l_c$. The full algorithm is explained by the equations (8) to (12). For this new method, we need to combine an algorithm to perform the observational update (OU) of the ensemble with the forward (ST) and backward ($\text{ST}^{-1}$) spectral transformations previously defined by Eqs. (2) and (3). Any observational update algorithm can be chosen provided that it allows localisation, for instance the SEEK observational update (Brasseur and Verron, 2006) or the ETKF observational update (Bishop et al., 2001). This localisation will be applied in our case in the spectral space

($\text{OU}_{\text{spectral}}$) or in the spatial space ($\text{OU}_{\text{spatial}}$) depending on the context.

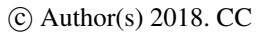



### 4.1.1 Spatial and spectral localisation

The analysis step is usually done in the spatial space with a spatial localisation (observational update $\mathrm{OU}_{\mathrm{spatial}}$) using spatial innovation. This step is applied to the prior ensemble anomaly, $\delta x_f^i$, with respect to prior ensemble mean (for member $i = 1, \cdots, m$) to obtain the updated ensemble $\delta x_a^i$. It corresponds to the correction applied to the prior ensemble during the assimilation step.

$$\delta x_f^i \rightarrow \mathrm{OU}_{\mathrm{spatial}} \rightarrow \delta x_a^i \tag{6}$$

Another approach is to apply the observational update in the spectral space with spectral localisation ($\mathrm{OU}_{\mathrm{spectral}}$) to the prior ensemble ($\delta x_f^i$) after transformation into the spectral space (ST). The spectral innovation is computed following Sect. 3.3.1. The resulting spectral analysis ($\delta x_a^i$ with superscript LS for large scales) is only available up to the scale $l_c$, for which the spectral transformation has been done.

$$\delta x_f^i \rightarrow \mathrm{ST} \rightarrow \mathrm{OU}_{\mathrm{spectral}} \rightarrow \mathrm{ST}^{-1}_{0 \rightarrow l_c} \rightarrow \delta x_a^{i,\mathrm{LS}} \tag{7}$$

### 4.1.2 Multiscale analysis : description of the algorithm

Multiscale analysis combines a spectral localisation for the large scales, and spatial localisation for the residual scales.

- *First step : spectral localisation for the large scales*

  Observational update with spectral localisation for the large scale part of the ensemble anomalies, as already described in the previous section, see Eq. (7).

  $$\delta x_f^i \rightarrow \mathrm{ST} \rightarrow \mathrm{OU}_{\mathrm{spectral}} \rightarrow \mathrm{ST}^{-1}_{0 \rightarrow l_c} \rightarrow \delta x_a^{i,\mathrm{LS}} \tag{8}$$

- *Second step : spatial localisation for the residual part*

  Extract $\delta x_f^{i,\mathrm{Res}}$ : the residual part of each anomaly of the prior ensemble :

  $$\delta x_f^i \rightarrow \mathrm{ST} \rightarrow \mathrm{ST}^{-1}_{0 \rightarrow l_c} \rightarrow \delta x_f^{i,\mathrm{LS}}$$
  $$\delta x_f^{i,\mathrm{Res}} = \delta x_f^i - \delta x_f^{i,\mathrm{LS}} \tag{9}$$

  Then, compute $\delta y^{\mathrm{Res}}$ : the residual part of the innovation, using the current best estimate of the large scale field at the observation points :

  $$\delta y^{\mathrm{Res}} = \delta y - \mathrm{H}\left(\frac{1}{N}\sum_{i=1}^{N} \delta x_a^{i,\mathrm{LS}}\right) \tag{10}$$

  Compute $\delta x_a^{i,\mathrm{Res}}$ : the residual part of the ensemble analysis increment, using the residual spatial innovation ($\delta y^{\mathrm{Res}}$) during the observational update in the spatial space with spatial localisation ($\mathrm{OU}_{\mathrm{spatial}}$). Spatial observational error has



to be estimated and can be smaller than those chosen for the spatial localisation only (see Eq. (6)) to get better results at each scale. Indeed, a part of the error has already been taken into account during the spectral localisation for the large scales.

$$\delta x_f^{i,\text{Res}} \to \text{OU}_{\text{spatial}} \to \delta x_a^{i,\text{Res}} \tag{11}$$

• *Third step : full spectrum*

Compute $\delta x_a^i$ : the final value of the ensemble analysis for the member $i$ as the sum of Eq. (8) and Eq. (11) :

$$\delta x_a^i = \delta x_a^{i,\text{LS}} + \delta x_a^{i,\text{Res}} \tag{12}$$

This analysis increment is directly comparable to the analysis increment obtained with the spatial localisation applied to the full field, see Eq. (6).

## 4.2 Comparison of localisation schemes

The relevance of implementing a multiscale analysis rather than the usual spatial localisation is only validated if spectral localisation better retrieves large scales patterns of the signal than spatial localisation. To verify the validity of this assumption, we perform two different analyses in the context of the twin experiments described in Sect. 2.2. The first analysis is carried out with a spatial localisation only following Eq. (6), hereinafter called spatial localisation, whereas the second analysis uses

a spectral localisation only following Eq. (7), hereinafter called spectral localisation. Spatial and spectral localisation radius have been optimised to obtain the best results in both experiments. The spatial localisation radius corresponds to a wavelength of spherical harmonic about $139\text{km}$ at the equator while those of spectral localisation is a rectangle of 3 in ordinate l-number and 1 in m-number. They have been deduced from the correlation ensemble (see for instance Fig. 2a and Fig. 2b for spatial localisation; and Fig. 5a and Fig. 5b for spectral localisation). The localisation radius are chosen small enough to avoid non

significant correlations. In order to evaluate these localisation algorithms, they are compared for the large scales. As justified later in Sect. 4.3, we choose to define large scales as the range of scales $l \in [0; 34]$.

Each scale of spatial and spectral analysis increments has to be as close as possible to the corresponding scale of the true anomaly. The large scale part of this spatial analysis increment ($\delta x_a^i$ of Eq. (6)) can be directly compared to the spectral analysis increment obtained from the large scale of the prior ensemble ($\delta x_a^{i,\text{LS}}$ of Eq. (7)). It can be extracted following Eq. (2) and

Eq. (3), to obtain the corresponding large scales of the spatial analysis increment :

$$\delta x_a^i \to \text{ST} \to \text{ST}^{-1}_{0 \to l_c} \to \delta x_a^{i,\text{LS}} \tag{13}$$

Simultaneously, the obtained spectral analysis increment (Eq. (7)) is back into the spatial space (applying $\text{ST}^{-1}$, following Eq. (3)) to be directly comparable to the large scales of the spatial analysis increment (Eq. (13)). Figures 6a and 6b show respectively the large scale of the mean ensemble of analysis increments obtained respectively with spatial localisation or

spectral localisation (see Sect. 4.1.1). Hence, they have to be as similar as possible to the large scale part of the true anomaly showed on Fig. 4c.



Spectral localisation recovers large scales much better than spatial localisation, see Fig. 6 vs. Fig. 4c. In all cases, the analysis increment is significant only where JASON data are available (see Fig. 1d). The analysis and the type of localisation thus have no significant impact on the north of the model domain. This result reinforces the idea of a multiscale analysis with spectral localisation for the large scale. It is now necessary to determine the critical scale, $l_c$, from which the spatial localisation will be

preferred.

## 4.3 Determining the critical scale

On average, spectral localisation only gives better results than spatial localisation for the large scales, but we need to check that this affirmation remains valid at each scale or that it exists a critical scale, $l_c$, from which this tendency is no more true. To determine $l_c$, a classic score is computed for the spatial localisation and the spectral localisation. It shows the improvement of

the RMSE after/before the analysis by averaging sums over the whole model domain :

$$\rho = \frac{\text{RMSE}_{\text{posterior}}}{\text{RMSE}_{\text{prior}}} = \frac{\sqrt{\overline{(<x_a> -x_{\text{true}})^2}}}{\sqrt{\overline{(<x_f> -x_{\text{true}})^2}}} \tag{14}$$

where $\overline{\cdot}$ is the mean over the domain. Each member of this equation is then computed for all specific degree $l$, following Eq. (2) and (3) with $l_{\min} = l_{\max} = l$. During an analysis step, the RMSE of the observed variable will be reduced. The ratio thus allows to evaluate the accuracy of the analysis for each scale. Figure 7 shows this score for each degree $l$ until 100 for the

spatial analysis only and until 60 for the spectral analysis only.

This gives a new point of view to evaluate the results of an analysis, giving the efficiency of the spatial or spectral analysis at each scale and no more only for the full field. Spatial localisation deals with all scales at the same time. The score is almost the same at each scale : around 0.8. In contrast, this score of the spectral localisation is very sensitive to the scale. It is almost up to twice smaller than the one of the spatial localisation for the large scales. It increases with the degree until being similar

and exceeding the spatial localisation score. The spectral observational error used for the spectral localisation has not been computed exactly from $l \approx 50$. This led to important values close to $l = 60$ which impacts the score shown in Fig. 7. However, if observational error has been computed exactly until larger degrees, the trend would follow a similar pattern.

Until around $l \approx 34$, spectral localisation further reduces the spatial RMSE than spatial localisation, which is consistent with the study of analysis increments in Fig. 6. While, for larger degrees $l$, this trend tends to reverse. The critical value, $l_c$, does not

need to be very precise for the multiscale analysis. Indeed, the scores of spatial and spectral localisations are close on a range of degrees (here around 30 and 50 for instance). A variation of a few degrees on $l_c$ will not have any major impact on the final results of the multiscale analysis. In this twin experiment and for all these reasons, the critical scale $l_c$ is now fixed to $l_c = 34$.





## 5   Evaluation of the multiscale observational update

The aim of this section is to evaluate the multiscale analysis and to compare it with spatial analysis, for the full spectrum but also at each scale. For that purpose, we did a multiscale analysis following the algorithm presented in the previous Sect. 4.1.2, with the critical scale $l_c = 34$. This experiment is hereinafter called spectral+spatial localisation in figures.

In Sect. 5.1, we demonstrate that multiscale analysis keeps the advantages of spectral and spatial localisations at each wavelength. This is done by studying the error of spatial RMSE for each scale and comparing the analysis increment for the multiscale analysis and the spatial localisation. In Sect. 5.2, we check the reliability of these updated ensembles by computing rank histograms in the spatial and also spectral spaces. In Sect. 5.3, we show that the spread of the updated ensembles obtained with the multiscale analysis decreases much more than those of the spatial localisation at large scales as well as for all scales.

In Sect. 5.4, we evaluate the impact of the multiscale analysis on the non observed variables (temperature and salinity), with a multivariate analysis. We show that on average multiscale analysis reduces much more their spatial RMSE than the spatial localisation for large scales, and similar errors at smaller scales.

### 5.1   Error reduction at each wavelength

On average, the updated ensemble produced with the multiscale analysis should better approach the true state than those

obtained with the spatial localisation only. To evaluate the efficiency of the multiscale analysis, the error has been computed in two ways : at each scale in the spectral space, following Eq. (14), and in the spatial space for the full spectrum, with a comparison of the analysis increments and the true state.

### 5.1.1   Spectral point of view : reduction of spatial RMSE for each scale

The previous score showing the evolution of the RMSE after/before the analysis on average on the model domain, following

Eq. (14) is now computed for the multiscale analysis. Figure 7 (black curve) shows this score computed for each scale until $l = 100$.

Multiscale analysis keeps the advantages of both localisations (spectral localisation in green and spatial localisation in blue). As expected, for the large scales $l \in [0; l_c]$ with $l_c = 34$, the multiscale analysis is much better than spatial localisation, and has the same order of magnitude as the spectral localisation. Indeed, the same spectral localisation with the same configuration

has been done for the multiscale analysis. For the residual scales, multiscale analysis allows to recover, as expected, similar results to the spatial localisation, especially for the smaller scales. Differences occur especially close to $l_c$ and result from the contribution of the spatial localisation to treat the residual scales.

### 5.1.2   Spatial global point of view : analysis increment

The analysis increments obtained with spatial localisation (Fig. 8a) or with multiscale analysis (spectral + spatial localisation,

Fig. 8b) can be directly compared to the full spectrum of the true anomaly shown in Fig. 1c. The multiscale analysis allows





to recover a part of the large scale pattern unlike the spatial localisation. It keeps advantages of the spatial localisation for the residual scales.

These analysis increments can be evaluated at each scales. Figures 6a and 6c show the large scales ($l \in [0; 34]$) of these analysis increments respectively for spatial localisation or multiscale analysis. They have been obtained from their respective

full fields (Fig. 8a and Fig. 8b) following Eq. (2) and Eq. (3). Small structures have been well removed from the full spectrum. They have to be closer as possible to the large scales of the true anomaly shown Fig. 4c, which have been extracted from its full field shown in Fig. 1c. Multiscale analysis and spectral localisation give similar results for the large scales and are better than the spatial localisation. This is consistent with observed reduction of spatial RMSE at each large scale, shown in Fig. 7.

## 5.2 Reliability of the updated ensemble

Updated ensemble should be reliable in spatial space but also in the spectral space. It means to check the coherence between the assumed probabilities and the observed statistics when the ensemble is compared to the verification data (the true state in our twin experiment, or observation in a real system). To check ensemble reliability, ranks are traditionally computed in the spatial space and summarised in a rank histogram. They show the distribution of observations with respect to the ensemble (Anderson, 1996; Talagrand et al., 1997). In our context of twin experiments, the prior ensemble is reliable by construction. Indeed, the

true state originates from the same ensemble simulation as the other members. The reliability of the updated ensemble will be evaluated by comparing the rank histogram of the updated ensemble with the rank histogram of the prior ensemble. Hence, a flat rank histogram indicates a reliable ensemble, whereas a U-shaped rank histogram indicates a lack of spread in the ensemble : the uncertainty is under-estimated (Anderson, 1996; Hamill, 2001). Alternatively, we propose a new point of view of these ranks, computing them in the spectral space. The interpretation of these new ranks has to corroborate the conclusions obtained

in the spatial space.

### 5.2.1 Spatial rank histograms

Rank histograms have been computed, with respect to the true state, from spatial maps limited to the JASON domain for the prior ensemble, the spatially updated ensemble and the multiscale updated ensemble. Figure 9b shows the rank histograms for the large scales ($l \in [0; l_c]$) for the same ensembles but also for the spectrally updated ensemble presented in the previous

section.

Ranks histograms show that all these updated ensembles can be considered as reliable as the prior ensemble, both for the full spectrum and for the large scales. Indeed, the prior ensemble looks somewhat under dispersed but can be considered reliable because the true member originates from the ensemble itself. The rank histograms of the updated ensembles are of the same order of magnitude as that of the prior ensemble. Thus, the small under-dispersion of the prior ensemble (which can only result

from the limited size of the sample) has not increased during the analysis step. These consistent ranks histograms confirm that the observational error have been properly evaluated.



### 5.2.2 A new point of view : ranks map in the spectral space

Reliability of all updated ensemble (spatial localisation only and multiscale analysis) is now tested for degrees $l \in [0;60]$ by calculating ranks in the spectral space with respect to the true state. Ranks are computed following the same procedure as in the spatial space but the members and the true state are previously transformed into the spectral space following Eq. (2). Figure 10 shows the maps of ranks in the spectral space for the prior ensemble, the spatially updated ensemble and the multiscale updated ensemble. The maps of ranks for the spectrally updated ensemble is not shown due to similar results to the multiscale updated ensemble. This new point of view allows to diagnose the behaviour of the system for each scale.

Ranks maps in the spectral space provide additional indication that all algorithms provides reliable updated ensembles. Observational error have been consistently evaluated. When degrees tend toward $l = m$, which corresponds to meridional signal, ranks are not all represented even for the prior ensemble. But, these spectral regions correspond to extremely weak standard deviation of the ensemble in the spectral space (see Fig. 3a) : there is no meridional signal in the prior ensemble . They do not have an important impact on the spatial field. For the other degrees, ranks show that the prior ensemble is reliable. The spatially updated ensemble and the multiscale updated ensemble also remain reliable even if the latter seems to be somewhat less dispersed.

### 5.3 Resolution of the updated ensemble

The spread, or variance, of the prior and the updated ensemble (with $N = 69$ members) has been computed to check the resolution of the updated ensemble. For instance, for the prior ensemble, $x_f$ :

$$\text{Spread} = \frac{1}{N} \sum_{i=1}^{N} \left( x_f^i - < x_f > \right)^2 \tag{15}$$

The reliability of the ensemble has been checked previously with the rank histograms. Then, the smaller the spread after the analysis, the better the analysis. Figures 11 and 12 show the spread of the prior ensemble, the updated ensemble after spatial localisation, and the updated ensemble after a multiscale analysis (spectral + spatial localisation), respectively for the full spectrum and after extraction of the large scales ($l \in [0, l_c]$, with $l_c = 34$).

The multiscale analysis allows to decrease the ensemble spread more than the spatial localisation. The spread is much more reduced along JASON tracks, see Fig. 11c. This decrease is especially important for the large scales, see. Fig. 12c. For the large scales, the spread of the updated ensemble by spectral localisation is not shown due to similar results to those of multiscale analysis. Thus, knowing that all these ensembles are reliable, the more efficient algorithm is the multiscale analysis because of it has further reduced the spread of the ensemble and is the closest to the the true state.

### 5.4 Multivariate analysis

Multivariate analysis consists in extending the observational update to non observed variables, like temperature and salinity, in the state vector during the analysis. The experimental setup remain the same. The aim is to evaluate the impact of the multiscale analysis on these non observed variables and to check that it does not introduce more error than the spatial localisation. These





errors could increase during the next forecast and cause some unrealistic values. For this purpose, we compute the score defined by Eq. (14) for each degree for the spatial localisation (spat) and for the multiscale analysis (spct + spat). Then, we compute for each level and each degree, the ratio of these scores, following Eq. (16).

$$\text{Ratio} = \frac{\rho_{\text{spat}}}{\rho_{\text{spct+spat}}} \tag{16}$$

Figure 13 shows these results for the temperature and salinity. Each depth of this figure thus correspond to the ratio of the blue and black curves of the Fig. 7, no more for the SSH but for the temperature or salinity instead of SSH.

On average, below and around the critical degree $l_c = 34$, the multiscale analysis further reduces the error as compared to the spatial localisation only. In a few cases, at basin scales, multiscale analysis appears to produce poorer results than spatial localisation. However, this effect is small as compared to the improvement made at the other depths and large scales. For

smaller scales, these two analysis give similar results. It is consistent with the fact that a similar spatial localisation is done for the both analyses and with the results obtained for SSH (see Fig. 7).

## 6    Conclusions

We have formulated and evaluated a multiscale analysis approach for ensemble ocean data assimilation that provides a better recovering of the large scales than the current spatial analysis with spatial localisation. It has been developed to be used in the

existing data assimilation system of Mercator Ocean used in the CMEMS project. This new scheme consists in performing a spectral analysis with spectral localisation for the large scales and a spatial analysis with spatial localisation for the residual scales.

The transformation to the spectral space and the spectral localisation provide consistent ensemble estimates of the state of the system (in the spectral space, or after backward transformation to the spatial space). In terms of accuracy, this spectral

localisation recovers the large scale structures better than the spatial localisation. For the large scales, spectral localisation yields lower errors than spatial localisation while keeping a reliable ensemble. Conversely, the spatial localisation is still preferable for the small scales.

This new spectral approach also gives a new point of view to diagnose the system. Traditional diagnostics as ensemble mean, spread, correlations structures, rank histograms, etc., gives information at each scales and no more only for the full field.

The multiscale analysis, which is a hybrid scheme combining spectral localisation for the large scales and spatial localisation for the residual scales, keeps the advantages of these two localisations. Consequently, it can significantly improve the current use of various ocean observing systems, particularly with regard to the large scale information contained in sparse distribution of observations as altimeters or ARGO floats.

The direct perspective of this study is to implement and test the method in the real CMEMS system developed at Mercator

Ocean. The target is (i) to check that the method can be applied without deep modification of the existing system, (ii) to evaluate the operational gain that is obtained by an improved control of the large scale signal, and (iii) to enhance the diagnostic of the system by evaluating the performance separately for each scale. Some data assimilation steps have already been successfully



carried out in the same context of our study (not shown). In the longer perspective, the implementation of this multiscale approach for ensembles might improve the CMEMS products of Mercator Ocean as the reanalysis which are used by a large scientific community.

*Acknowledgements.* This work was conducted as a contribution to the GLO-HR-ASSIM project, funded by the Copernicus Marine Environ-
5   ment Monitoring Service (CMEMS). CMEMS is implemented by Mercator Ocean International in the framework of a delegation agreement
with the European Union. Additional support to this study was also provided by the CNES/OSTST/MOMOMS project. The calculations
were performed using HPC resources from GENCI-IDRIS (Grant 2017-011279).



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





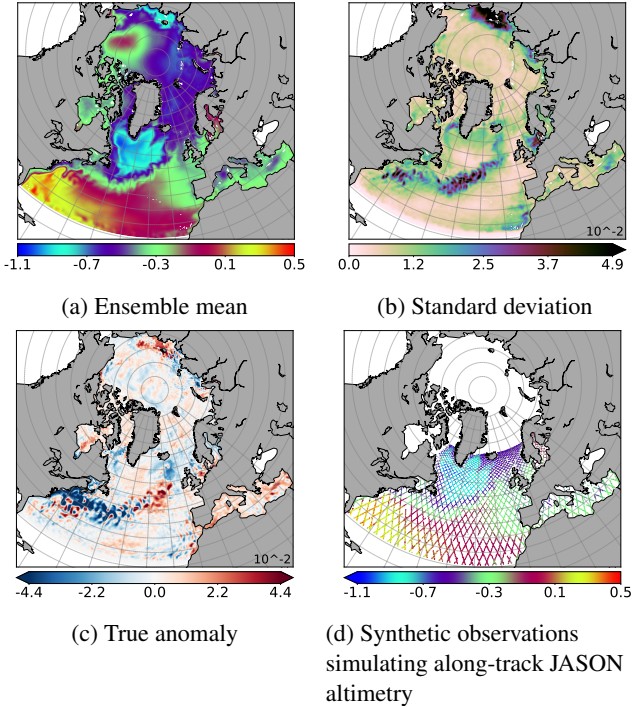

**Figure 1.** SSH (in m). Ensemble mean (a), standard deviation (b) of the prior ensemble ($N = 69$ members). The true anomaly (c) is defined as the difference between the true member (computed as an additional independent member of the ensemble) and the ensemble mean (a). Synthetic observations simulating along-track JASON altimetry (d) correspond to the true member along the track of JASON altimeter plus a noise, following Eq. (1).

| Source of uncertainty | pdf | Standard deviation | Corr. timescale | Laplacian filter |
|---|---|---|---|---|
| Equation of state | normal | 0.7/0.2 grid points | 8 days | 0 |
| Air-sea fluxes | gamma | 40% | 8 days | 3 |
| Ice strength | gamma | 40% | 8 days | 100 |
| Ice albedo | beta | 5% | 8 days | 100 |
| Ice drag coefficients | gamma | 10% | 8 days | 100 |

**Table 1.** Simulation of model uncertainties to perform the 70-member ensemble. It follows the working configuration used at Mercator Ocean to perform ensembles for research and development in the context of CMEMS systems.




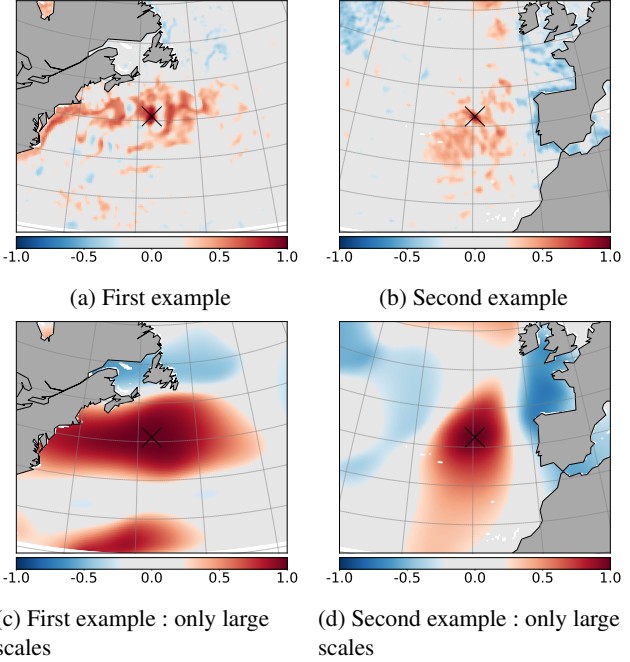

(a) First example  (b) Second example

(c) First example : only large scales  (d) Second example : only large scales

**Figure 2.** Two examples of ensemble correlation for the prior ensemble (SSH), according to two different reference points indicated by black crosses, computed for the full spectrum (top figures) and for the large scales (bottom figures, $l \in [0; l_c]$ with $l_c = 34$ following Eq. (2) and Eq. (3), which corresponds to a characteristic scale larger than $L \approx 187$km). Light grey colour corresponds to non significant values of ensemble correlations for an ensemble of $N = 69$ members (smaller in absolute value than 0.2367 with significance threshold at 95%).

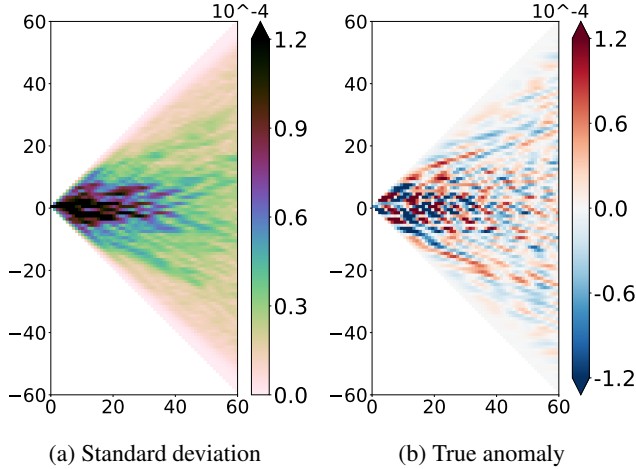

(a) Standard deviation  (b) True anomaly

**Figure 3.** Standard deviation (a) of the prior ensemble and true anomaly (b) in the spectral space (SSH), according to the degrees $l \in [0, 60]$ in abscissa and $m$ in ordinate, see Eq. (2), which corresponds to a characteristic scale larger than $L \approx 106$km




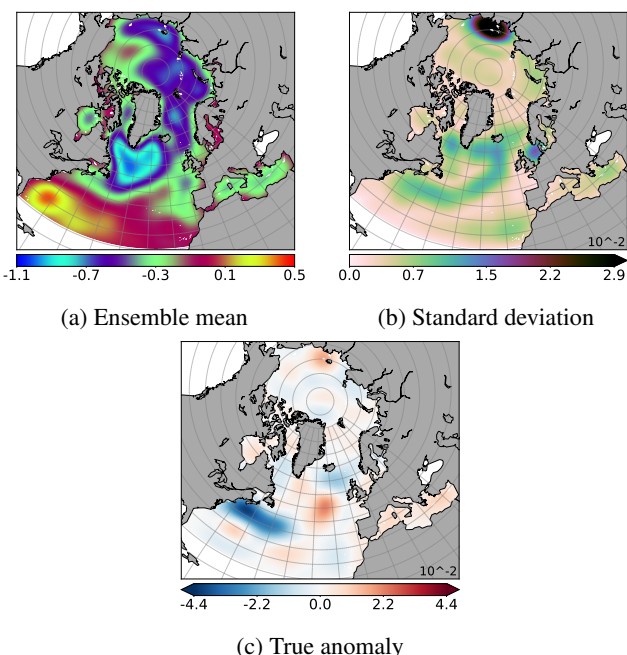

**Figure 4.** Ensemble mean (a), standard deviation (b) of the prior ensemble, and true anomaly (c) for the SSH (in m), after extraction of the large scales ($l \in [0; l_c]$ with $l_c = 34$, which corresponds to a characteristic scale larger than $L \approx 187$km, see Eq. (3)).

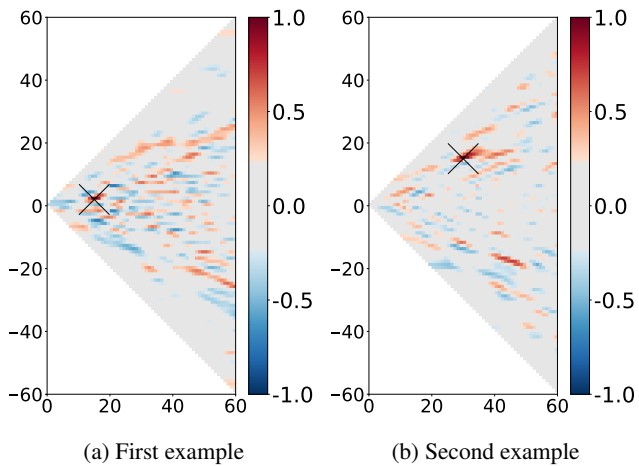

**Figure 5.** Two examples of ensemble correlation for the prior ensemble (SSH) in the spectral space, according to the degrees $l \in [0, 60]$ in abscissa and $m$ in ordinate, see Eq. (2). The two different reference points are indicated by black crosses. Light grey colour corresponds to non significant values of ensemble correlations for an ensemble of $N = 69$ members (smaller in absolute value than $0.2367$ with significance threshold at $95\%$).



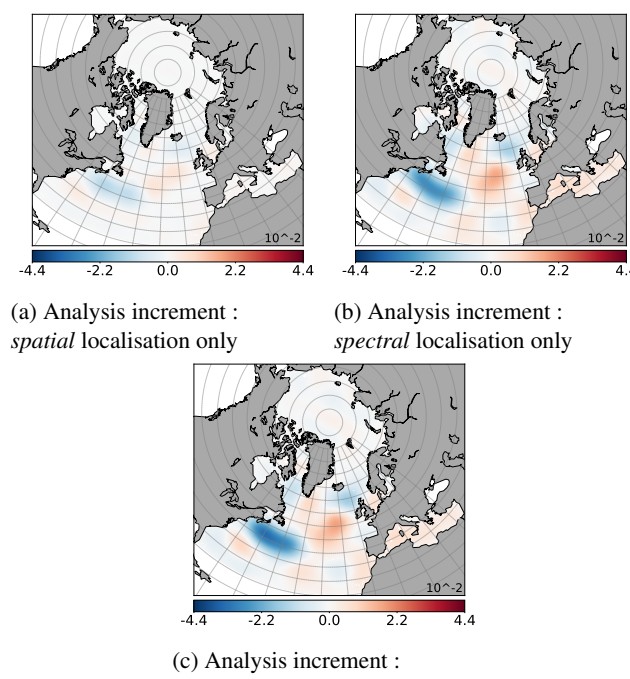

(a) Analysis increment :
*spatial* localisation only

(b) Analysis increment :
*spectral* localisation only

(c) Analysis increment :
*spectral + spatial* localisation

**Figure 6.** Ensemble mean of large scales part of the analysis increments (SSH in m), with $l \in [0, l_c]$, $l_c = 34$. (a) obtained after spatial localisation only and then filtered, see Eq. (6) and Eq. (3); (b) obtained after spectral localisation following Eq. (7); (c) obtained after multiscale analysis (spectral+spatial localisation, see Sect. 4.1.2). To be compared to the large scale true anomaly, Fig. 4c.

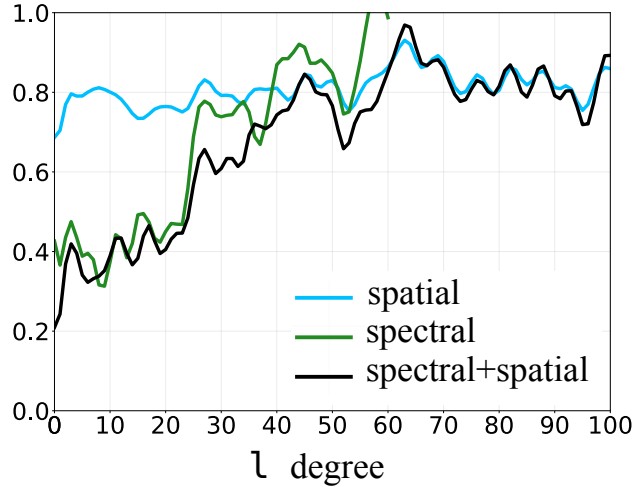

**Figure 7.** Reduction of spatial RMSE for each degree for the SSH, computed using Eq. (14). The blue curve (*spatial*) refers to the spatially updated ensemble, the green curve (*spectral*) to the spectrally updated ensemble, and the black curve (*spectral + spatial*) to the multiscale updated ensemble.




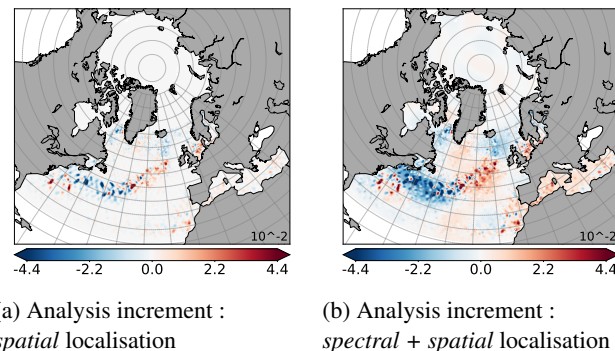

(a) Analysis increment :
*spatial* localisation

(b) Analysis increment :
*spectral* + *spatial* localisation

**Figure 8.** Same as Fig. 6a and Fig. 6c but keeping the full spectrum (no extraction of the large scales). (a) obtained after spatial localisation only, following Eq. (6); (b) obtained after multiscale analysis (spectral+spatial localisation) following Sect. 4.1.2. To be compared with Fig. 1c.

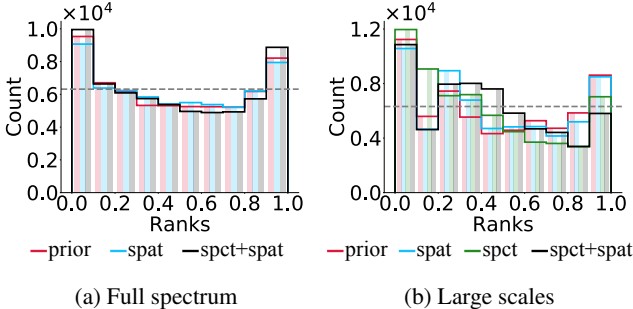

(a) Full spectrum

(b) Large scales

**Figure 9.** Spatial rank histogram on the JASON domain for the SSH. $prior$ (in red), $spat$ (in blue), $spct$ (in green) and $spct + spat$ (in black) correspond respectively to the prior ensemble, spatially updated ensemble, spectrally updated ensemble and to the multiscale updated ensemble. (a) Full spectrum; (b) After extraction of the large scales : $l \in [0; l_c]$ with $l_c = 34$.





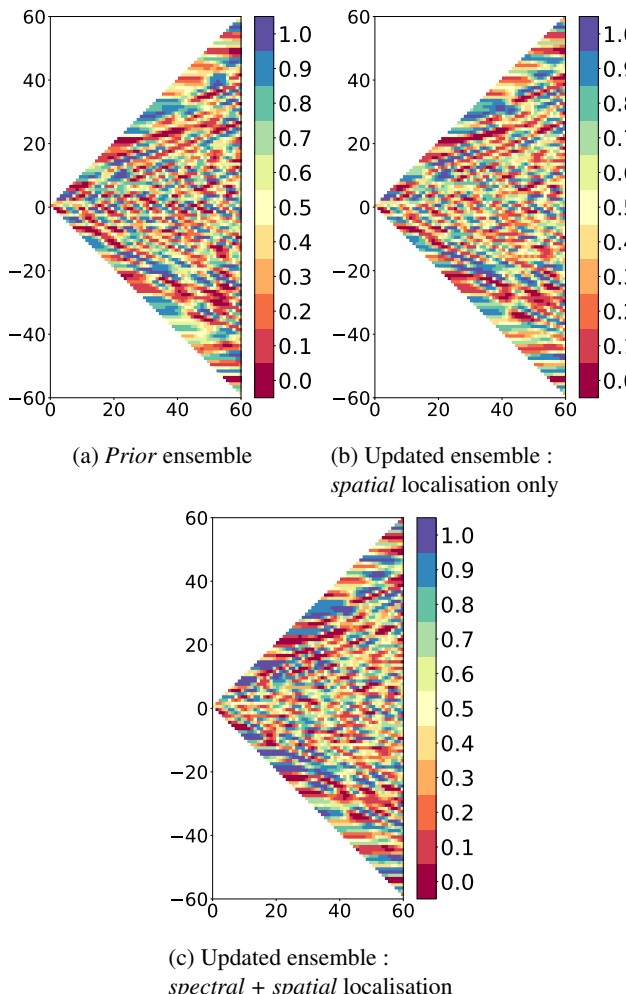

(a) *Prior* ensemble

(b) Updated ensemble :
*spatial* localisation only

(c) Updated ensemble :
*spectral* + *spatial* localisation

**Figure 10.** Maps of ranks in the spectral space for the SSH, according to the degrees $l \in [0, 60]$ in abscissa and $m$ in ordinate, see Eq. (2).
(a) Prior ensemble; (b) and (c) updated ensembles respectively obtained after spatial localisation only (see Eq. (6)) or after the multiscale
analysis (spectral+spatial localisation, see Sect. 4.1.2).





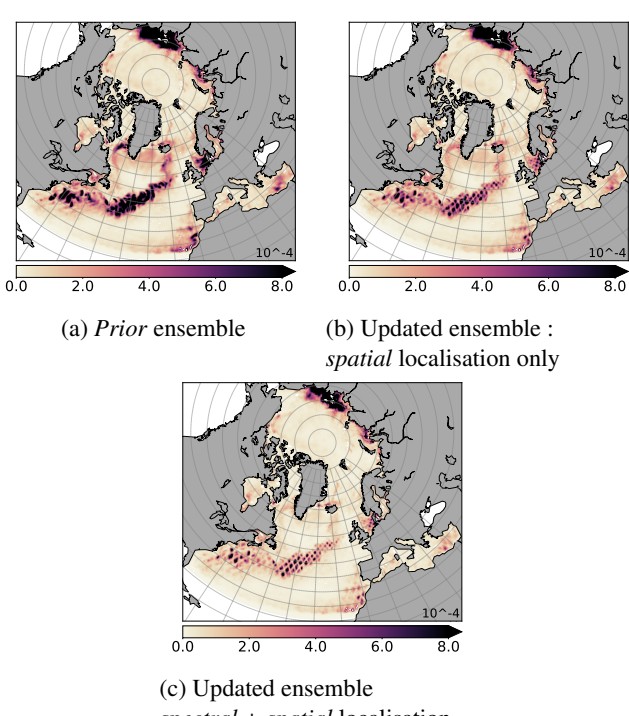

(a) *Prior* ensemble     (b) Updated ensemble :
*spatial* localisation only

(c) Updated ensemble
*spectral + spatial* localisation

**Figure 11.** Ensemble spread of the prior (a), the updated ensembles obtained with (a) the spatial localisation only or with (b) the multiscale analysis (spectral+spatial localisation), according to Eq. (15) (full spectrum) for the SSH.



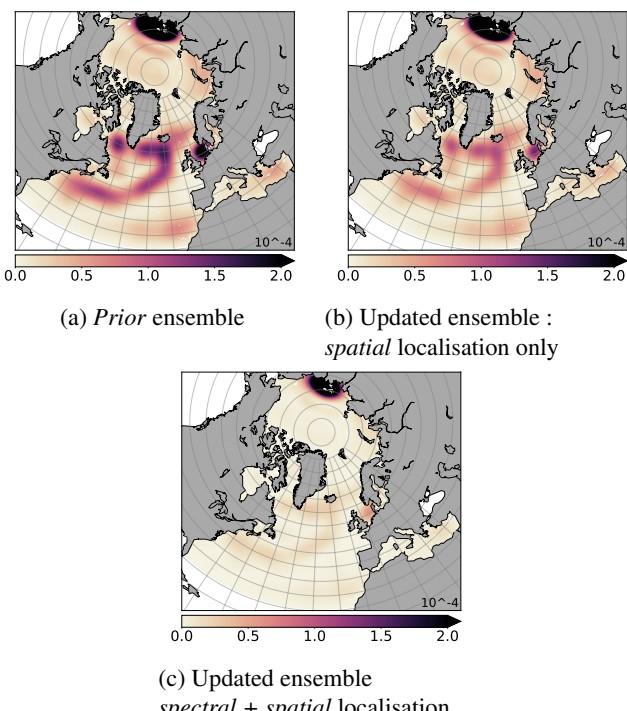

(a) *Prior* ensemble

(b) Updated ensemble :
*spatial* localisation only

(c) Updated ensemble
*spectral + spatial* localisation

**Figure 12.** Same as Fig. 11 but after extraction of the large scales ($l \in [0; l_c]$ with $l_c = 34$, see Eq. (2) and Eq. (3)).

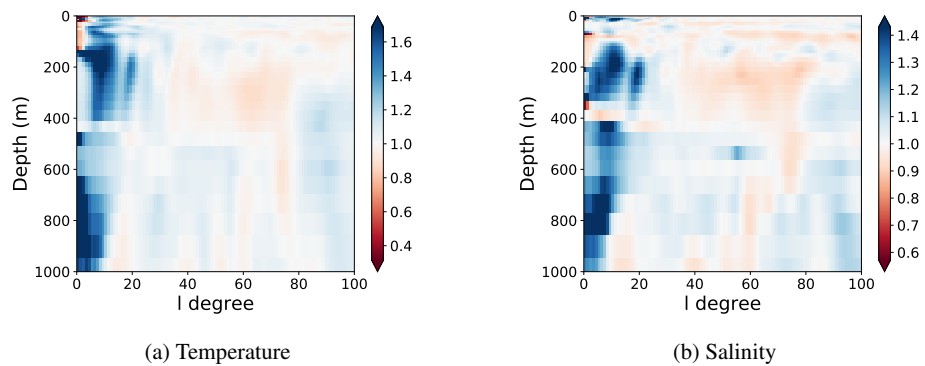

(a) Temperature

(b) Salinity

**Figure 13.** Improvement obtained by the multiscale analysis for temperature (a) and salinity (b). This improvement is measured by comparison to spatial localisation only using the ratio in Eq. (16). Blue (respectively red) colour corresponds to a better (respectively worse) correction of the error using the multiscale analysis as compared to spatial localisation only.