# Peer review of "A multiscale ocean data assimilation approach combining spatial and spectral localisation"

_Ocean Science, 2018_

## Referee Comment (RC1) · Anonymous Referee #1 · 14 Dec 2018

This paper presents a multiscale data assimilation method to take into account both large and small scales oceanic processes in an ocean operational system by spectral and spatial localisations. The paper is globally well written: most of the technical part is clear and the results are convincing.

My main questions are about the spectral transformation: 1) why the spherical harmonics transformation has been chosen? What is the strengths of this transformation compared to other spectral transformations?

2) It seems that after the spectral transformation, each term (corresponding to a given wavelength of a spherical harmonic) has particularlly a statistical sense, I just wonder if it also always has a physical interpretation, i.e. corresponds to an oceanic process at a specific scale. Even if so, do they really correspond to the scales observed in the

data?

3) It is not clear to me how the observation error standard deviation along each spherical harmonics is obtained in the twin experiment. More importantly, how it can be obtained in realistic case?

Minor questions or remarks:

1) In Fig. 7, why small difference exists between the black and the green curves before the critical scale while almost no difference between the blue and the black curves after the critical scale?

2) many small spelling mistakes, please check
* * *

---

## Referee Comment (RC2) · Anonymous Referee #2 · 28 Dec 2018

The manuscript is clearly written and presents a data assimilation approach focusing on more accurate retrievals of the large scale SSH components from the data.

The authors should better justify the use of eigenmodes of the laplacian on a sphere for scale separation. These basis functions are natural in atmospheric applications, but in the oceanic data assimation (into regional configurations in particular) it might be better to employ alternative expansions (e.g., laplacian eigenfunctions defined for the domain in use). Apart from being orthogonal, they have number of additional attractive properties, including spatial inhomogeneity of the supported scales and their implicit relation to model dynamics (e.g., tides).

Minor issues:

[Figure]

1) Referring to the impact spatial localization, I would rather say that large scale correlation structures are "heavily suppressed", but not "removed" or "not used" in the analysis associated with spatially localized covariance.

2) grammar issues (p.2: "spatial space", l.30, p.4: lines 7,14,20 etc..; caption to Fig. 11: description of panel c missing..). please correct

---

## Referee Comment (RC3) · Lemieux-Dudon (Referee) · 16 Jan 2019

This article focuses on the ensemble data assimilation systems which use the domain localization technique to prevent the contamination of the analysis with spurious long-range correlations due to ensemble error covariance matrix sampling errors. The authors propose a new data assimilation algorithm to solve the well-known drawbacks associated to the traditional localization techniques. Localization requires to define a decorrelation length (or radius of influence) which is known to be scale dependent. As a result a single decorrelation length cannot be suited for the wide range of scales represented in models and observations. Localization can also create noise especially when local analyses are superimposed. It also discards the true long-distance correlations (creating imbalance and loss of relevant information). This paper specifically

address this later issue. The objective of the authors is to improve the analysis of the large scales without worsening the small scale components of the signal in order to use the full constraint of the observing system at all scales. To separate the small and large scales, the authors apply a spectral transformation based on a spherical harmonics decomposition which enables to carry out two successive analyses : i) a first analysis is performed in the spectral domain with a spectral localization of the large scales components of the signal, ii) a second analysis is performed in the spatial domain with the traditional domain localization applied over the remaining smaller scales of the residual components of the signal. This approach called "multiscale observational update algorithm" is said to be computationally affordable because it avoids the small scale spectral transformation by using the multi-scale filtering technique.

This technique is tested in the framework of a twin experiment using synthetic satellite altimetry observations over a realistic regional configuration (ie, North Atlantic and Nordic Seas) at resolution of a quarter of degree. A prior ensemble of 70 members is simulated with NEMO v3.6 and the multi-scale analysis is implemented in the SAM2 data assimilation system (Mercator Ocean).

This article deserves to be published with minor changes. The state of art is properly introduced, the methodology is relevant and precisely described (eg, twin experiment), the results are commented and illustrated with numerous figures. Several verification methods are used to validate the proposed data assimilation algorithm (rank histograms, RMSE, ensemble spread before and after the analysis,..). Figure 7 is especially convincing in showing that the multi-scale analysis algorithm can improve the update of the large scale components without downgrading the small scales.

The objective of the authors appears to be reached when using synthetic SSH observations in a realistic regional configuration. I have few comments and questions which are listed below. The particular points for which I would recommend clarifications concern questions related to the transformation of the observations and the consistency analysis in the spectral space presented on Figure 10.

* p.2 line 28 : typography.

"compute removal of the between-scale covariances" : complete ?

* p.2 line 35 : methodology.

Can you explain the effect of neglecting the in between scale covariance ? (see related questions p.9 line 8).

* p.3 line 20 : suggestion.

This information was given several times and could be removed here : "This combines spectral localization for the large scales with spatial localization for the small scales"

* p.4 line 14 : typography.

uncertainties repeated twice

* p.4 line 29 : clarification appreciated.

The ensemble spread in the Gulf stream and Siberian Sea are too small/too large, respectively : "these characteristics do not affect the evaluation of the multiscale algorithm that is performed in this study". Can you explain why ?

* p.5 line 15 : language ?

"which is very close to whose used by the Ensemble Transform Kalman Filter (ETKF)"

* p.6 line 25 : suggested clarification.

"For a multivariate three-dimensional variable, this transformation can be applied to each vertical level of each model variable". Which physical parameter do you control and update? This question could be clarified when introducing the SEEK p.5.

* p.6 line 28 : minor comment.

"The reversible spectral transformation preserves the information for all degrees $l \leq l$ max" Information about lmax unclear at that stage : the spherical harmonics decomposition (ST) is not fully introduced yet.

* p.6 line 30 and p.7 line 1 : suggestion when referring to Figure 3a, 3b

You could explicitly mention that you are showing the f_lm coefficients of the spherical harmonics decomposition.

* p.7 line 7 : correction.

In Equation (3) (the inverse ST) since part of the original signal f(theta,phy) is very likely filtered out by applying the cut-off degrees lmin and lmax, I would suggest that you replace the name of the function f.

* p.7 line 16 : clarification.

Could you explain further what is the "scale separator" your refer to?

* p.8 line 1 : clarification / correction

The least square problem of equations (4) and (5) is set to identify the best spherical harmonic coefficients (ie, f_lm) so that the reconstructed observation field f(theta, phi) (here ssh) taken at the observation locations (ie, theta_k, phi_k) minimizes the distance to the observations (ie, fo_k) with possibly an additional regularization term (minimal norm) and/or some bogus observations. You mention that the least square problem is only used for observations which are not on a regular grid (else you would apply the ST directly). It is said that "p is the number of grid points of the domain" in equation (4). It seems to me that p is rather the observation space dimension.

* p.9 line 8 : clarification.

This comment is connected to my previous question about the impact of neglecting the in between scale covariance "p.2 line 35". You are mentioning here that the "correlations between very different scales are weak and should be neglected by the data assimilation scheme and reduced to zero". I am not fully convinced about this statement when looking at Figure 5a and 5b. You illustrate your statement showing the

cross-scales correlations for two "small" degree parameters (ie, large spatial scales) with the other spectral scales of the system (l in [0,60]). The cross-scales correlations "between very different scales" do not seem to be all negligible. Could you explain if these figures are meant to justify : i) the spectral scale separation algorithm (correlations between large and smaller spatial scales are said to be weak), ii) the need of a spectral localization for the large scales. What is the link with the next statement "For the same reasons, each localization window will contain a number of degrees of freedom sufficiently low to be controlled with an ensemble of moderate size"?

* p.11 line 28 : suggestion for Figure 6a, 6b and 6c

You mention that "Figures 6a and 6b show respectively the large scale of the mean ensemble of analysis increments obtained respectively with spatial localization or spectral localization (see Sect. 4.1.1). Hence, they have to be as similar as possible to the large scale part of the true anomaly showed on Fig. 4c". Comparison might be easier if you would directly plot the difference between the true anomaly and mean ensemble of analysis increments.

* p.13 line 29 : clarification

Regarding the results of Figure 8, did you apply zero bogus observations ?

* p.13 line 29: suggestion for Figure 8a and 8b.

Comparison might be easier for the reader if you directly plot the difference between the increment of analyses for the full spectrum and the True anomaly.

* p.15 line 8 : clarification needed for the rank histogram in the spectral space.

You mention that "Ranks maps in the spectral space provide additional indication that all algorithms provides reliable updated ensembles". You also say that "Figure 10 shows the maps of ranks in the spectral space for" the "ensemble". Rank histograms usually present the following information (similarly to Figure 9) along the x and y-axis: 1) the N+1 ranks extracted from the N ensemble members (ie, the N+1 probability

bins), 2) the counts of the verifying observation in each bins. On the "ranks map" of Figure 10, I cannot understand where are the counts of the synthetic observation used for verification (ie, the "true state") ? On such plot, it seems to me that one dimension is missing to measure the statistical reliability of the ensemble with respect to the verifying observation (ie, the counts). Is Figure 10 equivalent in a way to plotting several rank histograms for several spectral range ? What type of features you expect to see to confirm the statistical reliability of the ensemble (prior and updates) with such map of ranks ? Where are the counts ?

Finally, as stated in the other reviews there are small language issues which you should try to correct (I cannot help on that topic).

---

## Author Comment (AC1) · 21 Feb 2019

We thank referee #1 for his/her careful reading of the manuscript and for his/her appreciation for the work done in the paper. Comments are reproduced in bold italic to ease the reading. Text changes in the manuscript are in italics.

**Answer to general comments**

*This paper presents a multiscale data assimilation method to take into account both large and small scales oceanic processes in an ocean operational system by spectral and spatial localisations. The paper is globally well written: most of the technical part is clear and the results are convincing.*

[Figure]

*My main questions are about the spectral transformation: 1) why the spherical harmonics transformation has been chosen?  What is the strengths of this transformation compared to other spectral transformations?*

Yes, we agree with the reviewer that this point was not sufficiently explained in the paper. The following paragraph has been added in section 3.2 to better justify the choice of spherical harmonics to separate scales :

*"The use of spherical harmonics is not the most natural way to separate scales for fields that do not extend over the whole sphere.  In principle, it would for instance be better to use the eigenfunctions of the Laplacian operator defined for the model domain. They would account for the land barriers and would display a better relation to the system dynamics. However, they would also be much more expensive to compute than the spherical harmonics, and would need to be stored and then loaded each time they are needed to separate scales.  This is why we preferred using spherical harmonics in this study : they make the method numerically efficient and they are sufficient to obtain a relevant spectral decomposition of the input signal."*

*2) It seems that after the spectral transformation, each term (corresponding to a given wavelength of a spherical harmonic) has particularly a statistical sense, I just wonder if it also always has a physical interpretation, i.e. corresponds to an oceanic process at a specific scale.  Even if so, do they really correspond to the scales observed in the data ?*

Yes, they would have a clear interpretation if the domain extended over the whole sphere.  As explained above (in the paragraph now included in the paper), they are used here as an efficient practical way to separate scales.

***3) It is not clear to me how the observation error standard deviation along each spherical harmonics is obtained in the twin experiment. More importantly, how it can be obtained in realistic case?***

In the twin experiment, the observation error standard deviation along each spherical harmonics is obtained by transforming (i) the innovation vector and (ii) the misfit with respect to the true state, and by computing the RMS difference between these two transformed vectors. This sentence has been included in the paper in section 3.3.2 to improve the explanation :
*"It is then possible to evaluate the standard deviation of the observational error in the spectral domain by transforming (i) the innovation vector and (ii) the misfit with respect to the true state, and by computing the RMS difference between these two transformed vectors. More explicitly, this is done by computing the RMS between (...)"*

In a realistic case, the above method can directly be transposed by simulating observational error in model results, and by transforming the difference between the perturbed and unperturbed data. The standard deviation of the result is then an estimate of the observation error standard deviation along each spherical harmonics. The following sentence has been added in section 3.3.2 to clarify this point. *"In a realistic case, the above method can directly be transposed by simulating observational error in model results, and by transforming the difference between the perturbed and unperturbed data. The standard deviation of the result is then an estimate of the observation error standard deviation along each spherical harmonics."*

**Answer to minor questions or remarks**

*Minor questions or remarks :*
***1) In Fig. 7, why small difference exists between the black and the green curves***

[Figure]

*before the critical scale while almost no difference between the blue and the black curves after the critical scale?*

The analysis increment obtained with the spatial localisation can recreate a large scale field. This impacts the large scale of the total analysis increment. On the contrary, the analysis increment resulting from the spectral localisation will only impact the result obtained for the large scales, because the scales have been treated separately by the spectral localisation. The residual field may contain a small amount of large scales that have not been treated by spectral localisation. Thus, green and black curves before the critical scale (large scales) can be more different than blue and black curves after this critical scale (residual scales).

*2) many small spelling mistakes, please check*

We did our best to check the manuscript again, but the manuscript has not been revised by a native English speaker.

---

## Author Comment (AC2) · 21 Feb 2019

We thank referee #2 for his/her careful reading of the manuscript and for his/her appreciation for the work done in the paper. Comments are reproduced in bold italic to ease the reading. Text changes in the manuscript are in italics.

**Answer to general comments**

*The manuscript is clearly written and presents a data assimilation approach focusing on more accurate retrievals of the large scale SSH components from the data.*

[Figure]

*The authors should better justify the use of eigenmodes of the laplacian on a sphere for scale separation. These basis functions are natural in atmospheric applications, but in the oceanic data assimilation (into regional configurations in particular) it might be better to employ alternative expansions (e.g., laplacian eigenfunctions defined for the domain in use). Apart from being orthogonal, they have number of additional attractive properties, including spatial inhomogeneity of the supported scales and their implicit relation to model dynamics (e.g., tides).*

Yes, we agree with the reviewer that this point was not sufficiently explained in the paper. The following paragraph has been added in section 3.2 to better justify the choice of spherical harmonics to separate scales :
*"The use of spherical harmonics is not the most natural way to separate scales for fields that do not extend over the whole sphere. In principle, it would for instance be better to use the eigenfunctions of the Laplacian operator defined for the model domain. They would account for the land barriers and would display a better relation to the system dynamics. However, they would also be much more expensive to compute than the spherical harmonics, and would need to be stored and then loaded each time they are needed to separate scales. This is why we preferred using spherical harmonics in this study : they make the method numerically efficient and they are sufficient to obtain a relevant spectral decomposition of the input signal."*

**Answer to minor comments**

*1) Referring to the impact spatial localization, I would rather say that large scale correlation structures are "heavily suppressed", but not "removed" or "not used" in the analysis associated with spatially localized covariance.*

Yes, this is true. Long-range correlations are removed, but the something certainly remains from the large-scale correlation structure. This has been corrected in the paper.

*2) grammar issues (p.2: "spatial space", l.30, p.4: lines 7,14,20 etc..; caption to Fig. 11: description of panel c missing..). please correct*

Yes, thank you. This has been corrected. In particular, we changed "spatial space" and "spectral space" to "spatial domain" and "spectral domain" everywhere.

---

## Author Comment (AC3) · 21 Feb 2019

We thank Benedicte Lemieux-Dudon, referee #3, for her careful reading of the manuscript and for her appreciation for the work done in the paper. Comments are reproduced in bold italic to ease the reading. Text changes in the manuscript are in italics.

**Answer to general comments**

*This article focuses on the ensemble data assimilation systems which use the domain localization technique to prevent the contamination of the analysis with spurious long-range correlations due to ensemble error covariance matrix sampling errors. The authors propose a new data assimilation algorithm to solve*

*the well-known drawbacks associated to the traditional localization techniques. Localization requires to define a decorrelation length (or radius of influence) which is known to be scale dependent. As a result a single decorrelation length cannot be suited for the wide range of scales represented in models and observations. Localization can also create noise especially when local analyses are superimposed. It also discards the true long-distance correlations (creating imbalance and loss of relevant information). This paper specifically address this later issue. The objective of the authors is to improve the analysis of the large scales without worsening the small scale components of the signal in order to use the full constraint of the observing system at all scales. To separate the small and large scales, the authors apply a spectral transformation based on a spherical harmonics decomposition which enables to carry out two successive analyses : i) a first analysis is performed in the spectral domain with a spectral localization of the large scales components of the signal, ii) a second analysis is performed in the spatial domain with the traditional domain localization applied over the remaining smaller scales of the residual components of the signal. This approach called "multiscale observational update algorithm" is said to be computationally affordable because it avoids the small scale spectral transformation by using the multi-scale filtering technique.*

*This technique is tested in the framework of a twin experiment using synthetic satellite altimetry observations over a realistic regional configuration (ie, North Atlantic and Nordic Seas) at resolution of a quarter of degree. A prior ensemble of 70 members is simulated with NEMO v3.6 and the multi-scale analysis is implemented in the SAM2 data assimilation system (Mercator Ocean).*

*This article deserves to be published with minor changes. The state of art is properly introduced, the methodology is relevant and precisely described (eg, twin experiment), the results are commented and illustrated with numerous*

*figures. Several verification methods are used to validate the proposed data assimilation algorithm (rank histograms, RMSE, ensemble spread before and after the analysis,..). Figure 7 is especially convincing in showing that the multi-scale analysis algorithm can improve the update of the large scale components without downgrading the small scales.*

*The objective of the authors appears to be reached when using synthetic SSH observations in a realistic regional configuration. I have few comments and questions which are listed below. The particular points for which I would recommend clarifications concern questions related to the transformation of the observations and the consistency analysis in the spectral space presented on Figure 10.*

Thanks again for all suggestions made to improve the manuscript. We did our best to take them into account as explained in more details below.

**Answer to specific comments**

*\* p.2 line 28 : typography.*
*"compute removal of the between-scale covariances" : complete ?*

"compute removal" is replaced by *"complete removal"*.

*\* p.2 line 35 : methodology.*
*Can you explain the effect of neglecting the in-between scale covariance ? (see related questions p.9 line 8).*

This is indeed an important question. But it is difficult to provide general state-ments describing this effect. This can depend on every particular application. The

assumption is that the correlations between very different scales are weak and can be neglected. It is similar to the assumption made in spatial localisation where long-range correlations are neglected. In both cases, the effect is that information can be lost because of the localisation, and the importance of this effect depends on the magnitude of the correlation that have been neglected. The results of our example illustration tend to indicate that we loose less information with the multiscale approach than with spatial localisation alone. See our answer to the related question in p.9 to see how we tried to provide better explanations of this in the text of the paper.

*\* p.3 line 20 : suggestion.*
*This information was given several times and could be removed here : "This combines spectral localization for the large scales with spatial localization for the small scales"*

This sentence has suppressed.

*\* p.4 line 14 : typography.*
*uncertainties repeated twice*

"Uncertainties in air-sea fluxes uncertainties are" is replaced by *"Uncertainties in air-sea fluxes are"*.

*\* p.4 line 29 : clarification appreciated.*
*The ensemble spread in the Gulf stream and Siberian Sea are too small/too large, respectively : "these characteristics do not affect the evaluation of the multiscale algorithm that is performed in this study". Can you explain why ?*

The explanation was indeed a bit too short. The reason is that we are using twin experiments, so that the evaluation of the method does not depend much on the

realism of the model simulation, providing that the multiscale nature of the problem remains. To clarify, we modify the text as follows : *"However, since we are using twin experiments, the simulation does not need to be perfectly realistic to evaluate our approach (providing that the multiscale nature of the problem remains). These characteristics are thus not likely to affect the evaluation of the multiscale algorithm that is performed in this study."*

**\* p.5 line 15 : language ?**
**"which is very close to whose used by the Ensemble Transform Kalman Filter (ETKF)"**

This part of the sentence is replaced by *"which is similar to that used in the Ensemble Transform Kalman Filter (ETKF)"*.

**\* p.6 line 25 : suggested clarification.**
**"For a multivariate three-dimensional variable, this transformation can be applied to each vertical level of each model variable". Which physical parameter do you control and update? This question could be clarified when introducing the SEEK p.5.**

In principle, all variables of the state vectors should be updated by the observational update. In this paper however, we are not performing a full data assimilation experiment, and we are only evaluating the method by updating some of the model variables. It is first applied to sea surface height, which is the observed variable, and it is also applied to temperature and salinity (in section 5.4) to illustrate the application of the method to non-observed variables.
In order to clarify this point, we add in the text p.5, line 16 : *"In the assimilation system, the observational update is usually applied to all model variables. In this paper however, the effect of the multiscale approach will mainly be evaluated by*

*the update of SSH, which is the observed variable, and by the update of temperature and salinity to illustrate the application of the method to non-observed variables."*

*\* p.6 line 28 : minor comment.*
*"The reversible spectral transformation preserves the information for all degrees $l \leq l_{max}$" Information about $l_{max}$ unclear at that stage : the spherical harmonics decomposition (ST) is not fully introduced yet.*

The following sentences p.6 lines 28 and 29 :
"The reversible spectral transformation preserves the information for all degrees $l \leq l_{\max}$. Thus, the transformed fields contain the same information, until this degree, as that shown in the spatial space in Fig. 1." are replaced by :
*"This reversible spectral transformation preserves the information for each $l$ degree. The $f_{lm}$ coefficients of the spherical harmonics decomposition can be computed for each $l$ degree up to a selected degree $l = l_{\max}$. This transformed field contains the same information, until $l_{\max}$", as shown in the spatial domain in Fig. 1.".*

*\* p.6 line 30 and p.7 line 1 : suggestion when referring to Figure 3a, 3b*
*You could explicitly mention that you are showing the $f_{lm}$ coefficients of the spherical harmonics decomposition.*

The description of the figures 3a and 3b have been modified : p.6 line 30 to p.7 line 2.
*"The $f_{lm}$ coefficients of each member of the prior ensemble has been computed for the SSH until the degree $l_{\max} = 60$, which corresponds to a wavelength $\lambda \approx 667$ km and a characteristic length $L \approx 106$ km. Figure 3a shows the standard deviation of this prior ensemble in the spectral domain. Figure 3b shows the result of the spectral transformation applied to the true SSH anomaly, i.e. the $f_{lm}$ coefficients of the true SSH anomaly."*

*\* p.7 line 7 : correction.*
*In Equation (3) (the inverse ST) since part of the original signal $f(\theta, \phi)$ is very*
*likely filtered out by applying the cut-off degrees $l_{min}$ and $l_{max}$, I would suggest*
*that you replace the name of the function f.*

The function $f(\theta, \phi)$ is now called $f_{l_{\min}}^{l_{\max}}(\theta, \phi)$. We modified the text as follows
:
*"From the spectrum $f_{lm}$, the field $f_{l_{\min}}^{l_{\max}}(\theta, \phi)$ can be reconstructed using the inverse*
*transformation :*

$$\mathrm{ST}^{-1}_{l_{\min} \to l_{\max}}: \quad f_{l_{\min}}^{l_{\max}}(\theta, \phi) = \sum_{l=l_{\min}}^{l_{\max}} \sum_{m=-l}^{l} f_{lm} Y_{lm}(\theta, \phi) \tag{1}$$

*This inversion can be constrained to specific scales by choosing the values of $l_{\min}$ and*
*$l_{\max}$. The full field can be reconstructed since $f(\theta, \phi) = f_0^{\infty}(\theta, \phi)$. This is how the*
*method separates scales.*

*\* p.7 line 16 : clarification.*
*Could you explain further what is the "scale separator" your refer to?*

The scale separation operator corresponds to the forward and then backward
transformations. The scales are separated by choosing the $l_{\min}$ and $l_{\max}$ values for the
backward transformation, following Eq. (3).

*\* p.8 line 1 : clarification / correction*
*The least square problem of equations (4) and (5) is set to identify the best*
*spherical harmonic coefficients (ie, $f_{lm}$) so that the reconstructed observation*
*field $f(\theta, \phi)$ (here ssh) taken at the observation locations (ie, $\theta_k$, $\phi_k$) minimizes the*
*distance to the observations (ie, $f_k^o$) with possibly an additional regularization*

*term (minimal norm) and/or some bogus observations. You mention that the least square problem is only used for observations which are not on a regular grid (else you would apply the ST directly). It is said that "p is the number of grid points of the domain" in equation (4). It seems to me that p is rather the observation space dimension.*

Yes, this is true. $p$ is the size of the observation vector, including bogus observations. This has been corrected in the paper : *"where $p$ is the size of the observation vector (including bogus observations)"*

*\* p.9 line 8 : clarification.*
*This comment is connected to my previous question about the impact of neglecting the in between scale covariance "p.2 line 35". You are mentioning here that the "correlations between very different scales are weak and should be neglected by the data assimilation scheme and reduced to zero". I am not fully convinced about this statement when looking at Figure 5a and 5b. You illustrate your statement showing the cross-scales correlations for two "small" degree parameters (ie, large spatial scales) with the other spectral scales of the system (l in [0,60]). The cross-scales correlations "between very different scales" do not seem to be all negligible. Could you explain if these figures are meant to justify : i) the spectral scale separation algorithm (correlations between large and smaller spatial scales are said to be weak), ii) the need of a spectral localization for the large scales. What is the link with the next statement "For the same reasons, each localization window will contain a number of degrees of freedom sufficiently low to be controlled with an ensemble of moderate size"?*

Yes, we agree with the reviewer that this explanation was both too short and too optimistic. What the figures show is that most significant correlations are close to the reference scale. Most other cross-correlations are weak, which makes it reasonable

to treat the different scales separately during the analysis step. However, it is true that there are also significant correlations for remote scales, and that neglecting these correlations corresponds to loosing a potentially useful information. The only argument we can provide as a justification is that, in the spatial case (Fig. 2), there also exists significant correlations far from the observation, and that neglecting them also corresponds to loosing a potentially useful information. The same kind of hypothesis is thus made to apply spatial localisation, even if some useful statistical relationships may be neglected. The question is then : which of the two localisations better preserves the meaningful structures contained in the ensemble. The results of our example illustration tend to indicate that we can loose less information with the multiscale approach than with spatial localisation alone. We tried to clarify this point in the paper with the following text added in p.9.

*"Most correlations between very different scales are weak and might thus be neglected by data assimilation. This is the basic property allowing to introduce scale separation in the data assimilation scheme. However, it is true that there are also significant correlations for remote scales, and that neglecting these correlations corresponds to loosing a potentially useful information. This is however similar to what happens with spatial localisation : in Fig. 2, there also exists significant correlations far from the reference location. The question is then : which correlations is it better to neglect to preserve the meaningful structures contained in the ensemble. This is the question that we will try to elucidate with our example application."*

**\* p.11 line 28 : suggestion for Figure 6a, 6b and 6c**
**You mention that "Figures 6a and 6b show respectively the large scale of the mean ensemble of analysis increments obtained respectively with spatial localization or spectral localization (see Sect. 4.1.1). Hence, they have to be as similar as possible to the large scale part of the true anomaly showed on Fig. 4c". Comparison might be easier if you would directly plot the difference between the true anomaly and mean ensemble of analysis increments.**

Difference between the true anomaly and mean ensemble of analysis increments gives an information about how the analysis improves or not the ensemble according to the true state. However, knowledge of amplitudes of the true anomaly and these analysis increments is necessary to have a good physical understanding of what happens. This allows to check if the amplitude of the variable is physically accurate. For this reason, we prefer to keep our current figures.

*\* p.13 line 29 : clarification*
*Regarding the results of Figure 8, did you apply zero bogus observations ?*

Yes, we added zero bogus observations on the northern part of the domain, where there are no observations available from the SSH, see Fig. 1d. This has been clarified in the text of the paper.

*\* p.13 line 29: suggestion for Figure 8a and 8b.*
*Comparison might be easier for the reader if you directly plot the difference between the increment of analyses for the full spectrum and the True anomaly.*

For the same reasons as those relating to Figures 6a and 6b, we prefer to keep our current figures : knowledge of amplitudes of the true anomaly and these analysis increments is necessary to have a good physical understanding of what happens.

*\* p.15 line 8 : clarification needed for the rank histogram in the spectral space.*
*You mention that "Ranks maps in the spectral space provide additional indication that all algorithms provides reliable updated ensembles". You also say that "Figure 10 shows the maps of ranks in the spectral space for" the "ensemble". Rank histograms usually present the following information (similarly to Figure*

*9) along the x and y-axis: 1) the N+1 ranks extracted from the N ensemble members (ie, the N+1 probability bins), 2) the counts of the verifying observation in each bins. On the "ranks map" of Figure 10, I cannot understand where are the counts of the synthetic observation used for verification (ie, the "true state") ? On such plot, it seems to me that one dimension is missing to measure the statistical reliability of the ensemble with respect to the verifying observation (ie, the counts). Is Figure 10 equivalent in a way to plotting several rank histograms for several spectral range ? What type of features you expect to see to confirm the statistical reliability of the ensemble (prior and updates) with such map of ranks ? Where are the counts ?*

In this figure, the ranks are computed using to the true state in the spectral domain, i.e. for each spectral coordinate $(l, m)$. For a given rank, the count is obtained by counting the number of spectral coordinates $(l, m)$ belonging to that rank. It is possible to summarise these counts in a rank diagrams in the spectral domain like for the spatial domain. We decided to show the maps of ranks because it brings a new perspective and gives the same information. For a perfectly reliable ensemble, ranks would be evenly and randomly distributed over the entire spectral domain. The misunderstanding might come from the fact that is not the count that is displayed in the figures but the count normalized by the total number of data points to have numbers between 0 and 1. This was indeed not mentioned in the paper, and it is now clarified :
*"The ranks are computed for each spectral coordinate $(l, m)$, and have been normalized by the total number of data points to have numbers between 0 and 1. For a perfectly reliable ensemble, ranks would be evenly and randomly distributed over the entire spectral domain."*

*Finally, as stated in the other reviews there are small language issues which you should try to correct (I cannot help on that topic).*

We did our best to check the manuscript again, but the manuscript has not been revised by a native English speaker.
* * *